# Emerging Trends in Genetic Engineering of Microalgae for Commercial Applications

**DOI:** 10.3390/md20050285

**Published:** 2022-04-24

**Authors:** Samir B. Grama, Zhiyuan Liu, Jian Li

**Affiliations:** 1Laboratory of Natural Substances, Biomolecules and Biotechnological Applications, University of Oum El Bouaghi, Oum El Bouaghi 04000, Algeria; grama.samir@univ-oeb.dz; 2College of Marine Sciences, Hainan University, Haikou 570228, China; liuzhiyuan111@hainanu.edu.cn; 3College of Agricultural Sciences, Panzhihua University, Panzhihua 617000, China

**Keywords:** microalgae, genetic modification, genetic engineering, biotechnological production

## Abstract

Recently, microalgal biotechnology has received increasing interests in producing valuable, sustainable and environmentally friendly bioproducts. The development of economically viable production processes entails resolving certain limitations of microalgal biotechnology, and fast evolving genetic engineering technologies have emerged as new tools to overcome these limitations. This review provides a synopsis of recent progress, current trends and emerging approaches of genetic engineering of microalgae for commercial applications, including production of pharmaceutical protein, lipid, carotenoids and biohydrogen, etc. Photochemistry improvement in microalgae and CO_2_ sequestration by microalgae via genetic engineering were also discussed since these subjects are closely entangled with commercial production of the above mentioned products. Although genetic engineering of microalgae is proved to be very effective in boosting performance of production in laboratory conditions, only limited success was achieved to be applicable to industry so far. With genetic engineering technologies advancing rapidly and intensive investigations going on, more bioproducts are expected to be produced by genetically modified microalgae and even much more to be prospected.

## 1. Introduction

Microalgae (including cyanobacteria) are predominantly unicellular photosynthetic organisms which constitute the base of aquatic food webs. As an ancestor of plants with billions of years of evolutionary history, they distinctively adapted to extreme habitats and developed massive phylogenetic and biochemical diversity [1,2]. They have colonized almost all biotopes and been acclimated to severe environments, living in salt marshes, deserts or environments with very low light [3]. Not only do some microalgae tolerate hostile environmental conditions but also need these conditions to thrive. As a consequence of diurnal, seasonal, vertical and geographic variations plus fluctuations in nutrient availability, temperature, light and other factors, the distribution and metabolic activities of microalgae and the biomolecules they produce may be significantly more heterogeneous than previously believed [2].

In recent decades, microalgae emerged as new, attractive, promising and scalable platforms for the production of some biomolecules [4]. Among these biomolecules, some primary and secondary metabolites, such as carotenoids and proteins, were already commercialized as customer products, which intrigues further R&D to prospect more bioproducts from microalgae [5]. The efforts of developing these bioproducts from microalgae are underway, but with a number of constraints including: low product yield, slow growth rates, high cost of production facility, frequent contamination, high cost of harvesting, high energy consumption of cell lysis and complicated process of extraction of the desired metabolites [6]. The primary focus of the process development is to increase the yield of these biomolecules and the growth rates of microalgae. The evolution of genetic engineering technologies of microalgae progressed considerably for the past decades, and the research results obtained helped to boost the economic viability of commercial microalgal productions. This purpose of this study is to summarize the recent development and the emerging trends of molecular biotechnology applied to microalgae to increase the cells’ growth performance as well as to improve the synthesis rates of primary and secondary metabolites from this valuable group of organisms.

### 1.1. Phylogenetical and Biochemical Diversity of Microalgae

Microalgae could be prokaryotic or eukaryotic and are phylogenetically very diverse. The eukaryotic microalgae might be traced back 1.9 × 10^9^ years and are much younger than the cyanobacteria with 2.7 × 10^9^ years of phylogenetic history [1]. After the progenitor of the algae arose through an endosymbiosis with cyanobacterium, two evolutionary lines appeared simultaneously, namely green and red algae. These two lines acquired a number of various distinctive characteristics. Further groups of algae did not appear until much later through the secondary endosymbiosis, while the green and red algae are transformed into plastids in a eukaryotic host. This process gave rise to the *heterokont* algae, the *Dinoflagellates*, the *Cryptophytes* and the *Euglenida* [1]. Because of several biochemical and cellular disparities, two principal groups of green microalgae are identified: the *Chlorophyta* and the *Conjugaphyta*. While the second group is nearly five times larger than the *Chlorophyta*, none of the *Conjugaphyta* is yet applied for biotechnological engineering [1].

Algal diversity is very large and represents an almost unexploited natural resource. In accordance with some early assessments, there might be tens of thousands to millions of microalgae species, only a tiny fraction of which were isolated or described [2]. A more accurate number of estimation might be 72,500 species, which is still roughly twice as that of plants [7]. In recent decades, huge microalgal collections were generated by researchers from various countries. An example is the collection of freshwater microalgae from the Coimbra College (Coimbra, Portugal), considered to be among the largest in the world, with over 4000 different strains with more than 1000 species. The collection mirrors the broad array of microalgae ready to be used in various applications [8]. Microalgae are therefore a group that is not well studied in biotechnology terms. Of the great variety of microalgal species thought to exist, just some thousands of strains are preserved in collections around the world, with just some hundreds explored for chemical composition and only tens cultivated in manufacturing (tons per year) [9].

The wide variability of microalgae provides a large range of potential applications as source of feeding, stock of biomaterials and bioreactor of biotechnologically important molecules [8,10]. This diverse phylogeny is also expressed in a large biochemical diversity of pigments, photosynthetic storage products, mucilage materials, fatty acids, oils and hydrocarbons, sterols and secondary bioactive compounds, relating secondary metabolites [2,11]. Except for having a specific compound available that makes these organisms interesting, their diversity and the possibility of harvesting and cultivating under various conditions allows for using them as natural bioreactors for producing multiple chemicals through biorefinery or integrated processes [11].

### 1.2. Various Applications of Microalgae

The use of microalgae by humans was practiced since many years as food, feed, medicines and fertilizers. In the 14th century, the Aztecs harvested *Arthrospira*, formerly *spirulina*, a cyanobacteria in Lake Texcoco. They used tecuitlatl (cake made with *spirulina*) as a main dietary component. Most likely, the utilization of this cyanobacterium as food in Chad happened at the same period, or even earlier to the Kanem Empire (9th century) [9,12]. In a world with limited resources (energy, water and arable land, etc.) and intensifying anthropogenic pressure on the environment, the improvement of biotechnological processes to supply sustainable energy and renewable biomaterials from cleaner industrial processes constitutes a key challenge.

The genetic, phylogenetic and compositional nature of microalgal diversity is considerable, which makes them appealing for bioprospecting and the eventual industrial utilization of various biomolecules [13]. Nowadays, the principal industrial products from green microalgae are carotenoids and biomass for food, health and aquaculture [14]. These products are obtained from a restricted number of species; cyanobacterium from the *Arthrospira* genus constitutes 50% of world production, followed by green microalgae from the genera *Chlorella*, *Dunaliella*, *Haematococcus*, *Nannochloropsis* and the diatom *Odontella* [13]. Microalgae are usually selected based on both their growth properties and their aptitude to produce considerable amounts of specific metabolites. The efficiency of biomass production is a pivotal element of financial success in most of today’s commercial systems [14]. 

The production of such microalgal compounds depends on the variability of process and environmental conditions, which all have a direct effect on the value and quality of the target products. Quality control must therefore meet industry standards relating to toxicity and safety, the concentration of antioxidants and product characteristics relevant to marketability such as the taste and smell [14]. The market of microalgae applications remains in development, and the exploitation will spread into new areas. Given the huge microalgal diversity and modern technological progress of biotechnology, these microorganisms constitute so far major potential resources for novel products and applications [1]. Furthermore, as being broadly unexplored, the microalgae offer a major opportunity for discovery. The rediscovery rate (finding metabolites already described) should be superior to that of other groups of better-studied organisms.

### 1.3. Recent Development of Microalgal Biotechnology

During the current decades, the microalgal biotechnology acquired extensive and significant importance. Applications vary from biomass production for feeding to useful products for environmental applications [1]. Microalgal biotechnology is currently going through an unparalleled interest and investment worldwide [8]. For the last decade, researchers and industry developed various microalgal cultivation technologies that are in use today to produce biomass. Along with conventional fermentation reactors, two techniques are frequently applied in microalgae cultivation, specifically, outdoor and indoor production within photobioreactors. It is therefore important to develop and improve the diverse microalgae culture technologies to minimize production costs [11]. Growth rates significantly fluctuate between species and greatly depend on cultivation methods, especially the design of photobioreactors and the bioprocess conditions. In outdoor cultivation, extremophile species are preferred as they minimize the risk of contamination by competing organisms [14].

Recently, there has been a fast and significant progress in molecular engineering. As a result, the genetic technology applied to microalgae extended from the conventional process to systematic and synthetic regulation of metabolic pathways (Figure 1). Systems biology (including genomics, transcriptomics, proteomics and metabolomics) use two or more omics methods in order to define and to study a complex biological system. Synthetic biology aims to use standard biopartition as a basis for a new and fast biological implementation (e.g., transcriptome analysis combined to genome sequence might serve to predict promoter potency and operability). Metabolic engineering seeks to construct a strong host for the high production level of the chemical substance through genome editing and design of genetic circuits to redirect molecular flow of some metabolites [15]. 

## 2. Emerging Trends in Genetic Engineering of Microalgae for Commercial Applications

Genetic engineering has long been applied to microalgal technology for commercial applications for decades, and numerous reports are published [16,17]. It is not our intention to give an exhaustive review of relevant literature, and instead we try to give a synopsis of recent progress, current trends and emerging approaches in genetic engineering of microalgae for commercial applications, including production of pharmaceutical protein, lipid, carotenoids and biohydrogen, etc. Photochemistry improvement in microalgae and CO_2_ sequestration by genetic engineering of microalgae are also discussed since these subjects are closely entangled with commercial production of the above mentioned products. 

### 2.1. Genetic Engineering of Microalgae for Pharmaceutical Protein Production

Recombinant proteins are widely applied for industry, nutrition and medicine applications. When looking at the landscape of protein-based pharmaceuticals, glycoproteins represent the largest section of biologically derived drugs accepted by the European Medicines Agency, whereas CHO (Chinese hamster ovary) cells are the mostly used platform for this production [18]. A recombinant protein production using CHO cells can represent a several billion-dollar business. Apparently, the persistent increasing need for substantial quantities of biopharmaceuticals and their high and significant production cost in CHO cells, besides the factors related to virus contamination, nonetheless encouraged the improvement in contemporary alternative production systems [19].

The sector of industrial biology relies almost entirely on the utilization of heterotrophic platforms (bacteria, yeasts, mammalian and insect cells) for the production of pharmaceutical proteins, bioactive metabolites, and only recently microalgae attracted significant attention in representing an emerging alternative, which unveils multiple biotechnological advantages [18,19]. These photosynthetic eukaryotic microorganisms are safe and can easily grow in bioreactors at significant growing rates in a similar way as CHO cells. Microalgal systems are characterized by metabolic diversity, ease of genetic modification and may be cultivated less expensively [18,19,20]. Proteins expressed by eukaryotic microalgae can be modified after translation in a way similar to CHO cells, which facilitate the usage of microalgae recombinant proteins as pharmaceuticals, and the ability of microalgae to secret proteins into the growth media further simplifies the downstream processing [20].

Usually, recombinant proteins expression can be both in the nucleus and chloroplasts. Nuclear genome engineering permits protein secretion and posttranslational modifications including disulfide bond formation and glycosylation. Alternatively, chloroplast genome engineering ensures correct folding of the recombinant proteins, disulfide bonded and soluble so as to preserve biological activity, but they remain bound inside the plastids [18,19,20]. Moreover, chloroplast protein expression lacks post-translational modifications because of the bacterial origin of plastids [21]. The algal chloroplast allows higher expression efficiency compared to the nuclear genome because foreign genes can be focused on specific loci inside the chloroplast genome, leading to high-level and stable expression. Moreover, it is considered more attractive because it houses main metabolic pathways since the incorporation of transgenes happens through homologous recombination, and, thus, the insertion site can be simply defined [22,23]. Correspondingly, proteins with therapeutic value often undergo post-transcriptional and post-translational modifications, which are fundamental to the natural biological function. These include potential splicing, site-specific proteolysis, proper protein folding, glycosylation and disulfide bond establishment [24].

The chloroplast of *Chlamydomonas reinhardtii* can be considered as a new emerging platform to produce therapeutic proteins, due to the reduced cost of the production process, the absence of endotoxins in the cells and the availability of methods for foreign gene expression [23], while diatoms offer further advantages relative to some varieties of microalgae since gene expression within the nucleus allows the post-translational protein modifications and targeting to multiple intracellular sites [21].

Microalgae have become a potentially important platform for recombinant protein production (Table 1). Considerable efforts were performed to develop molecular tools to allow high-level transgene expression of both the nuclear and chloroplast genomes. Different recombinant proteins were successfully produced from microalgae, which include antibodies, immunotoxins, hormones, vaccines, industrial enzymes, nutraceuticals, supplements and other significant compounds, including the anti-cancer agent [18,19,20,21,22,23,24,25]. More than 50 various recombinant proteins were already effectively produced from microalgae [26]. Triton Health and Nutrition (San Diego, CA, USA) and Algenist (Manhattan Beach, CA, USA) are among the pioneering companies starting recombinant protein production from microalgae [27].

In the marine diatom *Thalassiosira pseudonana*, a bovine respiratory vaccine was successfully produced by using the protective antigen IbpA DR2 isolated from *Histophilus somni*. Using nuclear-based expression methodology, the recombinant protein synthesis rate was enhanced through synchronous targeting of both the cytoplasm and chloroplasts, which occurred during silicon limitation and growth arrest [21]. Additionally, the authors asserted that increasing light intensities by 2.5 times increased by 6-fold the expression levels to 1.2% of the total extractable protein, and CO_2_ supplementation increased protein yield per cell. By using an approach of foreign gene expression, the transgenic generation of endolysins in the *C. reinhardtii* chloroplast was reported by Stoffels et al. [23], and a stable accumulation of the endolysins was obtained up to ~1% of TSP (the total soluble protein), without any reduction in growth rate or biomass production. An antibacterial cell lysate activity was testified against various serotypes of *Streptococcus pneumoniae*, with resistance against penicillin and co-trimoxazole. Using the codon-optimized gene methodology, a stable transformant expressing an inhalant allergen in *C. reinhardtii*, namely Bet-v1, was generated by Hirschl et al., and allergen expression was achieved with yields ranging from 0.01 to 0.04% TSP [27]. The Bet-v1 exhibited the identical secondary structure as to that of the reference allergen derived by *Escherichia coli* and could bind to human IgE and murine IgG. In another study, using the same methodology, the successful nuclear transgenes’ expression of *Chlamydomonas* was also reported for HIV P24 antigen production [28]. By introducing the fully codon optimized gene versions of original wild-type strains, recombinant protein production increased up to 0.25% of overall protein content. The obtained transformants remained stable over time and culture cycles. With another method, other studies described a recipient strain and new vectors, which led to a codon optimized synthetic gene encoding human growth hormone (hGH) [22]. Synthetic algal promoters (*saps*) were designed to enhance nuclear gene expression in *C. reinhardtii*, and the results indicated that this synthetic promoter could reach greater expression levels for exogenous genes than the native promoter [29]. The authors demonstrated the usefulness of synthetic promoters to drive gene expression as a complementary method for the production of recombinant proteins. With another method, bacterial Tat export signal peptides were applied to translocate recombinant proteins within the thylakoid lumen of the *C. reinhardtii* chloroplast, and the result presented an innovative approach of using thylakoid lumen as a new compartment for producing recombinant proteins, which provided more protective environments for delicate proteins [30]. 

Enhancing rates of recombinant proteins’ synthesis and secretion by the microalgal cells is the other key challenge for achieving efficient industrial production. Recently, via a yellow fluorescent protein, Venus, as a reporter, the metalloprotease gametolysin signal sequence enabled secretion of recombinant Venus with yield of 1.3 mg L^−1^. This yield was improved up to 12-fold by the introduction of synthetic (SP) glycomodules within *C. reinhardtii*, leading to expression of this protein as fusion glycoproteins and reaching the highest secretion rate obtained to date for recombinant microalgae proteins [20]. The glycoprotein secretion did not decrease cell growth under the tested conditions. A case of the first and the successful cultivation of transgenic wall-deficient strains of *C. reinhardtii* at pilot scales (100 L) was reported, and constant production rates of cytochrome P450 and diterpene synthase in chloroplast were achieved [31]. A better growth in mixotrophic condition was observed compared to autotrophic growth regimes. Besides therapeutic proteins, several enzymes of industrial interests were also well generated in microalgae chloroplasts. However, most of desired metabolites are regulated by multiple enzymes instead of a monoenzymatic system, which requires the activation and overexpression of clusters of enzymes to obtain overproduction of the targeted molecules [32].

**Table 1 marinedrugs-20-00285-t001:** Examples of genetic engineering of microalgae for pharmaceutical protein production.

Microalgae Strain	Gene/Target Site	Approach	Results	References
*Chlamydomonas* *reinhardtii*	Endolysins Cpl-1 and Pal	Foreign gene expression	Total recombinant protein yield was ~1.3 mg/g algal dry weight	Stoffels et al. [23]
*Chlamydomonas* *reinhardtii*	Birch pollen allergen Bet v 1	Codon-optimized gene and stably integrated	Allergen expression with yields between 0.01 and 0.04% of TSP	Hirschl et al. [27]
*Thalassiosira* *pseudonana*	Antigen IbpA DR2	Nuclear-based expression	Increased recombinant protein by 1.2%	Davis et al. [21]
*Chlamydomonas* *reinhardtii*	HIV antigen P24	Codon-optimized	The yield of the recombinant protein increased up to 0.25% of the total cellular protein	Barahimipour et al. [28]
*Chlamydomonas reinhardtii* and *Chlorella vulgaris*	SARS-CoV-2 receptor binding domain (RBD) and basic fibroblast growth factor (bFGF)	Nuclear transformation	Up to 1.14 mg/g RBD and 1.61 ng/g FGF in *C. vulgaris* and 1.61 mg/g RBD and 1.025 ng/g FGF in *C. reinhardtii*	Malla et al. [33]
*Schizochytrium* sp.	Epitopes from tumor associated antigens	Cloning and ex-pression	BCB protein was expressed at levels up to 637 μg/g fresh weight	Hernández-Ramírez et al. [34]
*Haematococcus pluvialis*	antimicrobial peptide piscidin-4	Expression of codon-optimized	Confirmed that the antimicrobial peptide could be expressed from *H. pluvialis*	Wang et al. [35]
*Tetraselmis subcordiformis*	*rt*-PA	nuclear transformation	*rt*-PA was integrated, and the expression product was bioactive	Wu et al. [36]
*Fistulifera solaris*	*cox* gene	Cloning and ex-pression	The total content of Prostaglandins (PGs) was 1290.4 ng/g of dry cell weight	Maeda et al. [37]
*Schizochytrium* sp.	LTB:RAGE vaccine	Algevir system (inducible geminiviral vector)	Led to yields of up to 380 μg LTB:RAGE/g fresh weight	Ortega-Berlanga et al. [38]
*Schizochytrium* sp.	vaccine against Zika virus (ZIKV)	Algevir technology to express an antigenic protein	Antigen yields of up to 365 μg g^−1^ microalgae fresh weight	Márquez-Escobar et al. [39]
*Chlamydomonas reinhardtii*	Human interferon (IFN)	Cloning and ex-pression	IFN-α2a is expressed and it is functionally active as anticancer and antiviral agent	El-Ayouty et al. [40]
*Chlamydomonas reinhardtii*	PfCelTOS Antigen	Chloroplast expressed	Expressed recombinant PfCelTOS accumulates as a soluble, properly folded and functional protein	Shamriz et al. [41]
*Chlamydomonas* *reinhardtii*	Human growth hormone (hGH)	Codon-optimized and new vectors	0.5 mg hGH per liter of culture	Wannathong et al. [22]

### 2.2. Genetic Engineering of Microalgae for Lipid Production

Biochemical engineering approaches regulating the physiochemical conditions of cultivation, such as optimization of pH, salinity, temperature and nutrient levels, are widely investigated to produce lipid in microalgae. Recently, multi-omics techniques were increasingly employed to analyze microalgal lipid synthesis, and various approaches of genetic and metabolic engineering for increasing lipid production were applied to microalgae for strain selection and improvement (Table 2) [42].

The *Escherichia coli* acetyl-CoA synthetase gene (ACS, which catalyzes the conversion of acetate to acetyl-CoA) was introduced to *Schizochytrium* sp., a microalga of marine origin, in order to produce genetically modified ACS transformants. The results indicated that the overexpression of acetyl-CoA synthetase increased both the fatty acid fraction and the biomass by 11.3% and 29.9%, respectively [43]. Pyruvate dehydrogenase kinase (PDK) inhibits the pyruvate dehydrogenase complex (PDC) which catalyzes the oxidative decarboxylation of pyruvate. With another method using an antisense cDNA construct, which was cloned from the diatom *Phaeodactylum tricornutum*, a marine microalga, PtPDK antisense knockdown transgenic diatoms were generated. The concentration of neutral lipids in transgenic cells was increased by 82% without changing the fatty acid profiles [44]. The transgenic cells showed a slightly lower growth rate but similar cell size. Malic enzyme (ME) catalyzes oxidative decarboxylation of malate to pyruvate, producing pyruvate, NADH and CO_2_. Overexpression of malic enzyme in *P. tricornutum* enhanced lipid concentration up to 57.8%, an increase of 2.5-fold higher compared to the wild strain with no effect on the cell growth rates [45]. In another study, the overexpression of the malic enzyme isoform 2 (ME2) in *C. reinhardtii* PTS42 led to increased lipid synthesis rates up to 23.4%; the growth of transgenic lines was about 10% lower than the wild-type [46]. 

It is assumed that various microalgae induce the storage of TAGs during the day and reduce the storage overnight in order to meet cellular ATP demands. Therefore, inhibition of β-oxidation might avoid losing TAGs overnight, but it could result in reduced growth. This approach can constitute a successful tool to enhance the productivity of lipids from microalgae cells growing in closed photobioreactors with exogenous carbon sources under constant or diurnal light conditions [47]. Since both carbon chain length and the unsaturation degree may affect the qualities of biofuels derived from microalgae, a lot of efforts were carried out to adjust fatty acid profiles of microalgal species. Usually, microalgae species contain a fatty acid profile ranging between 14 and 20 carbon in length, which mostly are 16:1, 16:0 and 18:1 fatty acids. The most appropriate fatty acids used for biodiesel production should be 12:0 and 14:0. Transgenic overexpression of thioesterases could be employed to modify the chain length of fatty acids [48]. In another study, diacylglycerol acyltransferase (DGAT), the principal enzyme responsible for catalyzing the final step of biosynthesis of triacylglycerides (TAGs), was overexpressed in *P. tricornutum*, and it was found that a 35% increase of lipid synthesis could be achieved without compromising cellular growth capacity [48]. Using iterative metabolic engineering, *P. tricornutum* was engineered to accumulate the high value omega-3 long chain polyunsaturated fatty acid docosahexaenoicacid (DHA), which was performed by introducing the Δ5-elongase from the picoalga *Ostreococcus tauri* into the host cells. Expression of the heterologous elongase resulted in an eight-fold increase in DHA content [49]. In addition, the authors demonstrated for the first time the co-expression of two heterologous enzymatic activities (*OtElo5* and *OtD6*). By targeting other enzymes, the overexpression of diacylglycerol acyltransferase 2 from *Brassica napus* in *C. reinhardtii* resulted in higher production of polyunsaturated fatty acids, particularly α-linolenic acid, an important omega-3 fatty acid, by more than 12%, with no effect on the cell growth rates [50]. In another study, the overexpression of the same enzyme in *Neochloris oleoabundans* showed that the content of saturated fatty acid of C16:0 doubled up to 49%; C18:0 was decreased three times to 6%, and the content of total TAG was increased by 1.8 to 3.2 folds, compared to the control wild strain, with no significant difference in the biomass between the transformant and wild type [51]. Worthy of mentioning is that to enhance the content of lipids in cells, inhibiting the pathways which lead to other energy-rich storage compounds’ accumulation, such as starch, might also be a possible approach. A starchless mutant of *Chlorella pyrenoidosa*, referred as STL-PI, showed an increase of 20.4% more PUFA when cultured under optimum growth conditions and of 35.4% more PUFA when cultured under nitrogen-limited conditions, compared to the wild-type strain [8].

Transcriptomic analysis was applied to *N. oleoabundans* under nitrogen replete and deplete conditions, and the transcriptome was reassembled de novo, with the pertinent genes measured through the mapping transcriptome readings. Results showed that under nitrogen deprivation, carbon is distributed toward triglyceride formation, which was five times higher than under nitrogen replete conditions [52]. 

The availability of precursors seems to be among the main factors of regulating oil accumulation in microbes, and thus enhancing precursor availability for both primary metabolism and fatty acids synthesis, which might be an interesting strategy to improve lipid as well as biomass accumulation of microalgae simultaneously [53]. By examining a full cDNA transcript coding for diacylglycerol acyltransferase 1 (DGAT1), a gene coding for DGAT1 was identified in *Chlorella ellipsoidea*, and it was determined that it played a significant role in the accumulation of TAG. The study was the first case of DGAT1 analysis conducted in *Chlorella* [54]. 

The construction of metabolic pathways and assembled transcriptome using next generation DNA pyrosequencing technology was applied to *Dunaliella tertiolecta* marine microalga [55]. The transcripts and genes encoding key enzymes for biosynthesis and catabolism of fatty acids were successfully identified. Using another methodology, recent studies began to use transcription factor engineering (TFE) approaches to identify transcription factors (TFs) in microalgal species with the intention to boost lipid biosynthesis [56]. TFs control and regulate the specific target gene expression by binding to specific DNA motifs within cis elements of targeted genes and by reacting with the RNA polymerase to allow or prevent gene transcription [57]. The TFE approach is an innovative technology which enables regulation of multiple enzyme activities with single gene transformation, and, with the knowledge of TFs and their gene targets, the transcriptional control processes may be modified to manipulate the genes’ expression profiles which play roles in the production of target metabolites [56,57]. This approach could prove to be more efficient for generating global metabolic variations. A soybean transcription factor acting on the lipid levels from *Arabidopsis* was incorporated into *C. ellipsoidea*, and the result showed that 754 genes were highly upregulated, and 322 genes were downregulated for the transgenic strains, with increased gene expression levels and enhanced enzymatic activity of acetyl coenzyme A carboxylase observed in the transgenic strain. The resulting phenotype exhibited significantly enhanced levels of lipid by 52.9%, with no effect on the growth rate [58]. In the other study, PSR1 was identified in *C. reinhardtii* as TF regulators of TAG biosynthesis under lipid-inducing stress responses [59]. Recently, metabolically engineered *Synechocystis* sp. PCC 6803 (a freshwater cyanobacterium that can be cultivated in marine water) with an overexpression of acyl-ACP (an enzyme involved in the chemical activation of fatty acids’ synthesis) showed an increased lipid production by 5.4%, compared to the wild-type strain, with a slightly slower rate of growth than the wild-type [60]. 

The ability to produce gene knockouts is another approach for metabolic engineering to downregulate competitive pathways or to redirect fluxes towards the desirable product [61]. Using the CRISPR-Cas9 system, a knockout mutant of the phospholipase A2 in *C. reinhardtii* improved the lipid production up to 64.25% by preventing the degradation and hydrolysis of fatty acids from glycerophospholipids without a significant difference in the cell growth [62]. A CRISPRi based transcriptional silencing approach was used to downregulate the expression of the PEPC1 gene in *C. reinhardtii*, which was an essential gene regulation carbon flux, and it was found that PEPC1 downregulated stains exhibited lower chlorophyll content but high biomass concentration and lipid growth rates [63]. Recently, the RNP-mediated knock-out method that applied *C. reinhardtii* CC-4349 showed an increase in the oil productivity by 81% [64].

**Table 2 marinedrugs-20-00285-t002:** Examples of genetic engineering of microalgae for lipid synthesis.

Microalgae Strain	Gene/Target Site	Approach	Results	References
*Synechocystis* sp. PCC 6803	Acyl-ACP synthetase (aas)	Overexpression of aas	Increased lipid production by 5.4%	Eungrasamee et al. [60]
*Chlamydomonas* *reinhartdii*	Phospholipase A2 (*PLA2*)	Knock-out/CRISPR/Cas9	Improves the lipids’ production up to 64.25%	Shin et al. [62]
*Chlamydomonas* *reinhardtii* PTS42	Malic enzyme isoform 2 (*ME2*)	Overexpression	Increasing lipid rate up to 23.4%	Kim et al. [46]
*Chlamydomonas* *reinhardtii* CC400	*PEPC1*	Down regulation by CRISPRi/Cas9	Lipid (content and productivity of 28.5% DCW and 34.9 mg/L/day)	Kao and Ng. [63]
*Chlamydomonas* *reinhardtii*	*HpDGAT2D*	Heterologous expression	Increasing TAG content by ~1.4-fold	Cui et al. [64]
*Chlamydomonas* *reinhardtii* CC-4349	ZEP and AGP genes	CRISPR-Cas9 RNP-mediated knock-out method	Increased oil productivity by 81%	Song et al. [65]
*Nannochloropsis* *oceanica*	*AtDXS gene*	Engineering a control-knob gene	Lipid production increased by ~68.6% in nitrogen depletion and ~110.6% in high light	Han et al. [66]
*Phaeodactylum* *tricornutum.*	*GPAT* and *DGAT2 genes*	Overexpression	Total lipid content increased by 2.6-fold and reached up to 57.5% DCW	Zou et al. [67]
*Nannochloropsis* *salina*	*bZIP*	Overexpressed a bZIP TF, NsbZIP1	Lipid production increased by 50%	Kwon et al. [68]
*Scenedesmus* *obliqnus*	*Differential expression genes (DEGs)*	up-regulated genes	Lipid yield increased by 2.4 fold	Xi et al. [69]
*Phaeodactylum* *tricornutum*	*ptTES1*	Transcription activator-like effector nucleases (TALENs)	1.7-fold increase in TAG content	Hao et al. [70]
*Nannochloropsis* *oceanica*	*Transposome*	Insertion of a Transposome complex (mutagenesis)	Increased PUFA by 180% and EPA by 40%	Osorio et al. [71]
*Phaeodactylum* *tricornutum*	*PhyA*	Overexpression	Increased DHA by 12% and EPA by 18%	Pudney et al. [72]
*Synechocystis* sp.	*Acetyl-CoA carboxylase (ACC)*	Overexpression	Increased its lipid content by 3.6-fold	Fathy et al. [73]
*Chlamydomonas* *reinhardtii*	Diacylglycerol acyltransferase 2 (*DGAT*)	Heterologous expression	α-linolenic acid, an important omega-3 fatty acid, was improved by more than 12%	Ahmad et al. [51]

### 2.3. Genetic Engineering of Microalgae for Carotenoid Production 

Recently, microalgal technology received increasing interest to produce the valuable chemicals including carotenoids. Carotenoids are pigmented terpenoids which are synthesized by plants or microalgae as accessory pigments for a photosynthetic apparatus or as secondary metabolites in response to intracellular oxidative stress. Carotenoids are strong antioxidants with applications in the cosmetic, agro-alimentation, aquaculture and pharmaceutical industries [74,75]. Currently, most carotenoids are produced synthetically at industrial scales using petrochemical raw materials and cannot be used for human consumption, and the market calls for biologically produced carotenoids due to safety and environmental reasons [75]. The bioaccumulation of carotenoids in microalgae is a physiological response to counter the deleterious effects of intracellular oxidative stress caused by the reactive oxygen species (ROS), such as the superoxide anion or H_2_O_2_. The production of natural carotenoids by algal cell cultures holds particular promise because of its environmentally friendly and safe nature. 

Not surprisingly, the tools of molecular biology were applied to microalgae for carotenoid production (Table 3). In an effort to enhance lutein production in microalgae *Chlamydomonas* sp., carotenogenic gene expression profiles under highlight conditions were revealed, and it was found that extremely high light intensity led to upregulated transcription of zeaxanthin synthesis genes instead of that of lutein genes [76]. Using the nuclear transformation system with an anorflurazon-resistant phytoene desaturase (PDS) as a selection marker, the transformants overexpressing *PDS* allowed *Chlorella zofingiensis* to increase its overall carotenoid production by 32.1% with an additional increase of 54.1% astaxanthin; the transformants showed a similar growth rate to the wild-type [77]. The authors concluded that the *PDS* gene constituted a selectable dominant biomarker by which *C. zofingiensis* might be transformed to elevate the biosynthesis pathways of carotenoids. The metabolic engineering of *D. salina*, a saline microalga, was carried out to modify its biosynthetic pathways for ketocarotenoid production by the successful insertion of the β-carotene ketolase (*BKT*) gene from *Haematococcus pluvialis*. The results showed an increase in the productivity of both astaxanthin and canthaxanthin up to 3.5 and 1.9 µg g^−1^ DW, respectively [78]. With another method, by cloning and transforming the *BKT* gene into *H. pluvialis*, the overexpressed *BKT* gene resulted in increased total carotenoid and astaxanthin concentrations by 2–3-folds, whereas intermediates, such as echinenone and canthaxanthin, increased 8 to 10 times, compared to wild-type cells. Moreover, carotenogenic genes, such as the β-carotene hydroxylase gene (*bkh*), phytoene synthase (*psy*), lycopene cyclase (*lcy*) and phytoene desaturase (*pds*), are expressed at higher levels in transformed cells compared with the original strain of *H. pluvialis,* and the astaxanthin production was higher during the stress condition [79]. Using a comparative transcriptomic approach, the study by Huang et al. revealed that improved astaxanthin synthesis induced by a glucose addition was significantly associated with the upregulation of astaxanthin pathway genes and combined with downregulation of side pathway genes [80]. In addition, the transcriptome analysis showed that most genes related to the pathway of glycolysis were significantly upregulated during glucose supplementation. With another methodology, a mutant of *C. reinhardtii* with the zeaxanthin epoxidase gene being knocked out by using DNA-free CRISPR-Cas9 could increase both zeaxanthin content and productivity of zeaxanthin by 56- and 47-fold, respectively, without affecting lutein content [81]. The same xanthophyll pigment content was increased by 10–15% in *D. tertiolecta* mutants produced by the random mutagenesis technique with ethyl methanesulfonate (EMS) as a mutagenic agent; the specific growth rate was higher than that of the wild type under medium and high light conditions [82]. 

Recently, a chloroplast genetic engineer approach was applied to *H. pluvialis* for astaxanthin production. The coding sequence of endogenous phytoenedesaturase (*pds*) was codon-optimized and overexpressed within the chloroplast of *H. pluvialis*, and the results showed that astaxanthin accumulation increased 67% compared to the wild-type strain with no effect on the cell growth rates [83]. This is the first report of plastid transformation of a microalgae with its endogenous *pds* nuclear gene. Recently, using the upregulated expression method, *Haematococcus pluvialis* showed an increase in esterified astaxanthin (EAST) [84].

### 2.4. Genetic Engineering of Microalgae for Biohydrogen Production 

Molecular hydrogen (H_2_) may be generated by certain microalgae, which have an aptitude to reduce free protons into H_2_ through catalytic actions of specific enzymes, namely hydrogenases [92]. Hydrogenases (HydA) are enzymes encoded by the *hydA* gene in the nucleus. Its transcription and activity may only be enabled in anaerobic or sulfur-free conditions [93]. Hydrogenases’ activity can be attributed to the metal ions at their active sites, and the (FeFe) and (NiFe) hydrogenases can reversibly reduce protons to H_2_ [47]. These two types of enzymes are phylogenetically different, and it is of interest to note that (FeFe) hydrogenases were reported exclusively in eukaryotic microalgae, and only (NiFe) hydrogenases were reported in cyanobacteria. The (FeFe) hydrogenases which are present in several green microalgae may efficiently interact with the transport chain of photosynthetic electrons within the ferredoxin, thus directly generating H_2_ produced by water oxidation [47]. The main limitation of H_2_ production is that the hydrogenases are highly intolerant to O_2_ released from the photosynthesis process. Moreover, the availability of reducing agents, such as ferredoxin and NADPH, constitute an added bottleneck since they are also consumed by other pathways such as respiration [94]. 

Various methodologies for H_2_ production from microalgae are reported, and plenty of investigations are underway. Melis et al. used sulfur deprivation as an effective and innovative approach, by partially inhibiting the PSII activities and balancing the photosynthesis and respiration abilities inside a cell to decrease photosynthetic activity and to establish an anoxic environment within the culture, under which conditions H_2_ production occurs [94]. This way, a two-step method was established for H_2_ production, with the first step to accumulate biomass under normal conditions through photosynthesis and with the second step under anoxic conditions to produce H_2_ [95,96]. Genetic engineering was applied to increase H_2_ photoproduction efficiency in green microalgae in combination with biochemical process strategies, and various methodologies and approaches are in development to improve H_2_ production capacity. To overexpress hydrogenase (hydA), the engineered hydrogenase gene (*hydAc*) was introduced into plasmids and induced by the promoters that could function under aerobic conditions. The hydrogen production from transgenic *Chlorella* sp. DT using over-expressed homologous HydA had a 7- to 10-fold improvement in H_2_ production than the original strain under similar conditions [93]. 

Another study applying the antisense RNA construct in *C. reinhardtii* showed that the attenuation of sulphate transport gene (*SulP*) expression also led to hydrogen accumulation and decreased PSII activity [97]. The authors concluded that *anti-SulP* strains are promising in reducing the supply of sulfate to the chloroplast, resulting in a downregulation of H_2_O-oxidation and O_2_-evolution activity. By using an approach of RNA interference (RNAi), mutant *Chlamydomonas* strains, which had reduced activity in ferredoxin-NADP+ reductase (FNR), were generated, and results showed as well that the fnr-RNAi strains achieved 2.5-fold higher hydrogen photoproduction rates than wild-type strains under conditions of sulfur-deprivation. In addition, the rates of cell growth of fnr-RNAi strains in TAP medium were almost comparable to the wild-type [98]. The results provide new information about FNR in regulating H_2_ metabolism. With another method, the genetic elimination of metabolic pathways consuming reducing agents NAD(P)H by knocking out lactate dehydrogenase in *Synechococcus* 7002 resulted in a five-fold increase in H_2_ production compared to wild-type cells, with no significant difference in the rates of photoautotrophic growth and aerobic respiration [99]. 

Other authors demonstrated that truncation of light harvesting antennas can improve H_2_ photoproduction efficiency in algal mass culture under high light conditions [100]. The results of the study showed that the obtained strain presents almost four times more H_2_ at 285 µEm^−2^ s^−1^ and over six times more at 350 µEm^−2^ s^−1^. To generate anaerobic cellular conditions, an OEC (the oxygen-evolving complex) dysfunctional mutant of *Chlorella* sp. strain DT was created by knocking down PsbO, a protein subunit for O_2_ evolution. The result showed that, under semi-aerobic conditions, PsbO-knockdown mutants might produce hydrogen photobiologically ten times more than the wild-type [101].

Another route to facilitate H_2_ biosynthesis is to sequestrate O_2_ produced by photosynthesis. Both genes of the soybean leghemoglobin protein, which sequesters oxygen within nitrogen-fixing root nodules, and the ferrochelatase of *Bradyrhizobium japonicum* were de novo synthesized and transformed into chloroplasts of *C. reinhardtii* to test the effects on cellular H_2_ synthesis, and the results showed that a 22% increase in H_2_ production and an overall increase of 134% in O_2_ uptake with about 12.5% inhibited of the growth rate than the wild strain were achieved [102]. This study confirmed the potential of the utilization of leghemoglobins for H_2_ production in green algae.

Recently, transcriptomic analysis powered by next-generation sequencing technologies was also applied to biohydrogen production from microalgae. The study of transcriptomic modifications through hydrogen generation from *C. reinhardtii* by microarrays and RNA-seq showed a novel insight regarding the control and rearrangement of the internal metabolism of cells. The result characterized all major processes that provided energy and reduction equivalents during hydrogen production. Furthermore, *C. reinhardtii* exhibited a significant increase in the transcription of model genes in charge of the stress response and detoxification of oxygen radicals [103]. These results provide clear evidence that both methods, RNA-seq and microarray, were able to supply additional information to clarify further the regulation of genes during hydrogen production. Within this context, identification of common transcripts among different microalgae and their differential expression during H_2_ production was reported as a promising research route [104]. In this study, 156 million reads generated from seven samples were used for de novo assembly after data trimming. The results show that more contigs were expressed differentially in the period of early and higher H_2_ photoproduction, and fewer contigs were differentially expressed as H_2_ photoproduction rates decreased. The authors described a method for analyzing RNA-Seq data without reference genome sequence information, which can be applied to other non-sequenced micro-organisms. 

Comparative proteomics were investigated in *C. reinhardtii* to identify the proteins involved in anaerobic adaptation. A total of 2315 proteins were identified using spectral counting analysis, and, among those, 606 proteins were found in chloroplasts, many of which were proteins of fermentation metabolism. Comparative quantitative analyses were carried out on the basis of the localized chloroplastic proteins with stable isotope labeled amino acids and revealed novel targets for additional investigations, such as proteins with protective roles from ROS and proteins of unidentified roles in anaerobiosis. Moreover, it was demonstrated that some of these proteins were upregulated during transcription under anaerobic conditions [105]. Quantitative data identified several novel proteins of unknown function that were produced under anaerobic conditions. A number of these proteins were found to be regulated at the transcript level which may bring insights for the engineering of hydrogen-producing alga strains. The anaerobic production of H_2_ from microalgae might be broadly applied to electricity generating with a similar conversion efficiency as a counterpart operated with chemically pure H_2_, which was exampled by coupled hydrogen production using sulfur-deprived *C. sorokiniana* cells with a Proton Exchange Membrane Fuel Cell (PEMFC) [106]. There is a growing interest in the new research areas such as integrated biosystems to produce H_2_ in mixed-cultures and direct utilization of residual algal biomass as a primary input for hydrogen production by a consecutive anaerobic digestion stage. Interdisciplinary approaches, such as nanotechnology and electrochemistry, immobilization of cells on novel materials and utilization of wastewater resources as a convenient and renewable source of nutrients for H_2_ production [65], were also reported [107]. Recently, Khanna et al. reported a novel methodology regarding apo-hydrogenase activation via insertion of a synthetic cofactor in vivo, and the potential to create catalytically active “semi-synthetic” hydrogenases in living cells was demonstrated [108]. This new method offers a potential to screen different pertinent host organisms easily, without the requirement of co-expressing the maturation machinery efficiently. In addition, the introduction of synthetically modified cofactors can generate enzymes with novel catalytic characteristics and facilitate spectroscopic studies of the enzyme under in vivo conditions. Finally, the ability to activate the enzyme instantaneously provides a rare tool for gain-of-function studies, allowing detailed analyses of the effect of an active (FeFe) hydrogenase. This new approach may result in easy identification of different relevant host organisms, can generate enzymes with novel catalytic properties and might provide a specious tool to develop comprehensive knowledge about the impact of operating hydrogenase (FeFe) on the condition of the host cells. Mutagenesis using atmospheric and room temperature plasma (ARTP) was applied to *C. reinhardtii* for improvement of H_2_ production from the microalga, and it was reported that H_2_ production was raised to 5.2 times compared to the wild-type strain. The reduction in antenna sizes found in mutants might be the reason for the enhancement of H_2_ production because antenna size reduction could increase the total photosynthesis efficiency of microalgal cells [109].

### 2.5. Genetic Engineering of Microalgae for CO_2_ Sequestration

CO_2_ sequestration by microalgae and photochemistry improvement of microalgae via genetic engineering are also discussed in this review since these subjects were closely entangled with commercial production of microalgal metabolites. The relationships between metabolites’ production and these two subjects are schematically represented in Figure 2. 

Coupling CO_2_ sequestration and commercial production of microalgal biomass might be a viable option for developing microalgal biotechnologies. Recently, significant advances were made to enhance engineering techniques for increasing CO_2_ fixation in microalgae. The Rubisco (primary CO_2_-fixing enzyme), which catalyzes oxygenation or carboxylation of ribulose-1,5-bisphosphate during Calvin cycles, constitutes the most common protein on the earth and exists ubiquitously in various photosynthetic organisms, and there has been great interest in modifying its activity for many years [110]. Cyanobacterial Rubisco has a relatively low affinity with CO_2_, compared with that of microalgae or plants. Because of this, cyanobacteria developed a CO_2_ concentrating mechanism (CCM) which involves two carbon-fixing enzymes, Rubisco and carbonic anhydrases, contained in carboxysomes. It is believed that these micro storage compartments enhance CO_2_ levels surrounding Rubisco to compete the association of Rubisco with another substrate, O_2_ [94,111]. There is nowadays an emerging and strong interest in engineered phototrophic organisms to fix more carbon using this enzyme by selecting genes specific to carbonic anhydrase and Rubisco or engineering metabolic pathways [94,110,111]. In a study, the inactivation of carbon concentration machinery through transcript knockdown of a cytosolic carbonic anhydrase (CA2) resulted in ~45%, ~30% and ~40% elevation in the photosynthetic oxygen evolution rate, growth rate and biomass accumulation under 5% CO_2_ in the *Nannochloropsis* sp., a marine microalgae [112]. In this way, inactivation of CCM may generate hyper-CO_2_-assimilating and autonomously containable industrial microalgae for flue-gas-based oil production. An important approach to increase carbon assimilation through the Calvin cycle is to enhance Rubisco activity. Atsumi et al. genetically modified *Synechococcus elongatus* PCC7942 to overexpress Rubisco for production of both isobutyraldehyde and isobutanol using CO_2_ [113]. With this Rubisco overexpression approach, there was a doubling of productivity with respect to the wild-type strain. The engineered strain produced isobutyraldehyde at a higher rate in comparison to those reported for ethanol, hydrogen or lipid production. In addition, several studies contributed to identify how the enzyme is folded and assembled and also to comprehend the main functions of the enzyme and its subunits involved in catalysis. Efforts were also tried to combine different sub-units from various organisms, each with different affinities with CO_2_, through protein engineering. However, the final enzyme might not be functional because of structural differences which led to, e.g., wrong assembly [110].

The development and emergence of recent synthetic biology facilitated and allowed the design, the synthesis and the introduction of biochemical pathways (a synthetic pathway) in vivo. In another study, a synthetic photorespiratory bypass on the grounds of the 3-hydroxypropionate bi-cycle (one of the six known CO_2_ fixation cycles in nature) was engineered using *S. elongatus* sp. PCC 7942 to re-assimilate glyoxylate, which resulted from the photorespiration pathway. The heterologously expressed cycle was engineered to perform as both a photorespiratory bypass and an added CO_2_-fixing pathway, complementing the Calvin–Benson cycle. The bypass might allow cells to avoid any carbon and nitrogen pert during the photorespiration and thus increase the rate of CO_2_ fixation [114]. As an effective and innovative approach, this study is among the first successful efforts to express a synthetic CO_2_-fixing photorespiratory bypass in a photoautotrophic organism. Another option of gene manipulation is to use the approach of small regulatory RNAs (sRNAs) that operate as transcriptional and post-transcriptional controllers of gene activity in organisms. A study of small RNAs with differential expression levels in the wild-type of *Synechocystis* sp. strain PCC6803 under high-light intensity revealed a sRNA (*RblR*) fully complemented to its targeting rbcL gene encoding the large chain of Rubisco. The *RblR* (+)/(−) mutants showed that RblR operated as a positive controller of *rbcL* within different stress conditions. Furthermore, the suppression of *RblR* had a negative effect on carbon assimilation, suggesting a regulatory function of the *RblR* for CO_2_ fixation [115]. A mechanism was suggested in which the interaction of RblR and its complementary mRNA masks RNase E cleavage sites and protects the target mRNA from degradation. 

### 2.6. Genetic Engineering of Microalgae for Photochemistry Optimization 

Microalgae are recognized as large and important groups of micro-organisms to be exploited for commercial and other applications, and the photosynthesis process efficiency of microalgae might be improved by genetic engineering more easily than that of terrestrial plants since they are single-cell organisms and proliferate similar to other micro-organisms. Algal cells have developed great light-harvesting complexes (LHC) to maximize light absorption. Under highlight conditions, excess energy absorbed is dissipated by heat as well as fluorescence quenching through the LHC. Excess energy that cannot be dissipated usually induces the ROS production [116]. The intensity of light at which a strain of microalga achieves its maximum growth rate, corresponding to the maximum efficiency of photosynthesis, is generally around 200 to 400 µmol photons m^−2^s^−1^ for the majority of the microalgal species. With the light intensities above maximum, efficiency of photosynthesis decreases, and the growth rates level off. This phenomenon is called photolimitation, and with the light intensity further increasing, the growth rates of microalgae decrease, which lead to photoinhibition [47]. Photolimitation and photoinhibition may consequently or concurrently occur during growth cultivation, which limits both growth and photosynthetic efficiencies. To overcome such a problem, several strategies were implemented, such as modifying the light harvesting complex to reduce light harvesting capacity, improving light utilization by changing the light composition and decreasing non-photochemical quenching [117].

Several approaches were implemented to increase photosynthesis efficiency and/or to reduce the impact of photoinhibition on microalgal growth. Recent development has been focusing on decreasing the chlorophyll antennae sizes and reducing numbers of light harvesting complexes to diminish the assimilation of sunlight [118]. Using the mutagenesis approach of the tla1 gene (gene responsible for regulating the chloroplast antenna size in green microalgae) resulted in a partially truncated chlorophyll antenna, with a functional chlorophyll antenna size of photosystem I and photosystem II being reduced about 50% and 65%, respectively, compared with the wild-type, and the mutant showed higher efficiency of solar conversion and growing rates than the wild-type [119]. The authors concluded that the partially truncated chlorophyll of the tla1 mutant antenna prevents the overabsorption of incident sunlight, attenuates the unnecessary dissipation of the absorbed irradiation and decreases the rather severe gradient of light and mutual shading of the cells as well as allowing a more uniform illumination of the cells in mass culture. In another approach, RNAi technology was used to downregulate the gene expression of the whole light-harvesting antenna complexes (LHC). The resulted mutant Stm3LR3 exhibited a significant decrease in LHC mRNA and protein content although the function of chlorophyll and pigment synthesis was maintained. In addition, reduced rates of fluorescence and increased light transmissions by 290% were also observed during the cultivation together with decreased photoinhibition sensitivity. The cell growth was improved under high light conditions possibly due to improved light penetration, and oxidative photodamage of PSII was observed to be less serious than the wild-type [120].

Using a translation repression approach, engineered translation repressor NAB1 of LHC was introduced into *C. reinhardtii* to decrease LHC antenna sizes, and LHC antenna sizes were found to be reduced by 10% to 17% with improved photosynthesis rates, increased by 50% under highlight conditions in mass cultivation of the microalgae [121]. Moreover, the cultures achieved higher densities when they were cultivated in large-scale bioreactors. In order to test the applicability of the Truncated Light-harvesting Antenna (TLA)-concept in cyanobacteria, transformants with smaller sizes of the phycobilisome (PBS) light-harvesting antenna were generated by the suppression of operon CPC in *Synechocystis* sp. PCC 6803 which encoded for various proteins needed for the fixation of the PBS peripheral phycocyanin rods. The results showed a substantial increase in photosynthesis efficiency by doubling the photosynthesis saturation intensity and 1.57 times greater biomass accumulation in relation to the native strain [122]. In another work, chloroplast signal recognition particle 54 gene (*CpSRP54*) was identified as the lesion causing the truncated light-harvesting antenna phenotype in *C. reinhardtii*, and it was proposed that a mutation of *CpSRP54* might be used to generate tla (truncated light harvesting antenna) mutants in microalgae with a significantly smaller size of photosystem antenna and thus improving conversion rates of solar energy and enhancing photosynthetic efficiency in a high-density culture, leading to the greater growing rate compared to the wild-type [123]. It has long been found that, NAB, a cytosolic RNA-binding protein in *C. reinhardtii*, serves as the key factor to regulate the translation of PSII-associated light-harvesting proteins (LHCBMs) after transcription, and the expression of NAB1 can be induced by CO_2_ limitation [124]. The accumulation of NAB1 decreases the functional PSII antenna size in the wild-type, which prevents states of damaging overexcitation of the PSII to happen, as occurs in a mutant without NAB1. Results of recent research showed that translation regulation as a recently identified long-term response to extended CO_2_-limitation facilitated LHCII state transitions including rapid reactions to PSII over-excitation, and the NAB1 was identified as a key factor of regulation linking short- and long-term photoacclimatation reactions, adjusting PSII excitation pressure [125]. This study provides significant new insights into the multilevel regulation of light harvesting by demonstrating how short- and long-term responses are coordinated in a regulatory circuit based on intracellular communication between the chloroplast, nucleus and cytosol. 

By using the RNAi technology method to downregulate almost the entire LHC antenna system (0.1–26%, subunit specific) [120], analysis of the thylakoid ultrastructure of the mutant of *C. reinhardtii* indicated that this LHC down-regulation reduces interactions between adjacent thylakoid membrane pairs within the granal stacks consistent with the loss of PSII supercomplexes. In addition, the resulted mutant showed reduced fluorescence losses, increased photosynthetic quantum yield, increased resistance to photoinhibition and a faster growth rate at elevated light levels. By using an approach of overexpression genes, photosynthesis efficiency of *Synechocystis* sp. PCC6803 was increased by overexpression of the flv3 gene, which coded NAD(P)H oxygen oxidoreductase, and the reason might be attributed to enhanced photosynthetic electron transport and supply of ATP by quicker regeneration of NADP+ in the recombinant strain [126]. The authors reported that the cell growth and glycogen production were increased. In addition, the increase in O_2_ evolution and ATP accumulation indicates enhancement of the alternative electron flow (AEF). In another approach to increase photosynthesis efficiency, it was found that D1 site-directed mutants of *Chlamydomonas* could reach and keep more than 2-fold higher O_2_ evolution rates even under the very high intensity light up to 900 µmol m^−2^ s^−1^ [127]. The mixotrophic growth rates showed mostly the same trends among the different mutants compared to the wild-type. Although plenty of research have been carried out to improve the photochemistry of microalgae, and a lot of mutants with higher photosynthetic efficiency were reported under laboratory conditions in the last decades, the experiments at pilot scales are not reported so far. The photosynthetic system is composed by large complexes of closely interactive proteins, and any major breakthroughs would require the simultaneous engineering of many of these proteins that are maintained throughout the evolutionary history of photosynthetic systems [116].

## 3. Opportunities, Challenges and Prospects

According to recent literature reviews, the whole genomic sequencing of more than 40 microalgal species have been completed, and much more are underway [117]. Recent approaches of next-generation sequencing technologies including sequencing-by-synthesis, single-molecule real-time sequencing and pyrosequencing found wide applications in the functional genomic research of microalgae, such as profiling of mRNAs and small RNAs, recognizing transcription factor regions, genome annotation, detection of aberrant transcription, mutation mapping and polymorphic identifying of noncoding RNA [128,129]. All these techniques were implemented to facilitate sequencing of several microalgal species and offered novel and rapid ways for genome-wide characterization. In parallel, several bioinformatics methodologies were developed to assemble, map and evaluate huge quantities of relatively or extremely short nucleotide sequences generated with next-generation sequencing data [130]. Emerging comparative genomics allow the investigation of the variability and operability of algal genes by intra- and inter-species comparisons [131]. Three recent reviews summarize the multi-omics analysis, recent genome-scale metabolic simulation, genetic engineering and evolutionary engineering of microalgae previously reported and might be able to provide detailed information and knowledge in this fields [15,132,133].

Three organelles available to microalgae, the nucleus, mitochondria and chloroplast, could be subjected to genetic engineering for commercial production of microalgal biomass or metabolites. Plenty of investigations were reported to be able to improve targeted metabolite production by engineering single genes that are crucial for the synthesis of those metabolites. The alternative approach is transcriptional engineering (TE), which regulates a wide range of genes involving multiple metabolic pathways simultaneously [134]. TFs control specific target gene expression by binding specific DNA sequences with cis elements of the gene target and by acting with the RNA polymerase to allow or disallow the transcription. Through knowledge of TFs and possible genetic targets, the transcriptional monitoring mechanism can be engineered to regulate the expressed genes, thus modifying relevant metabolic pathways [57].

Advanced genome-editing techniques for microalgae such as zinc finger nuclease (ZFN), meganuclease (MN), TALEN and CRISPR/Cas9 techniques recently emerged as powerful tools for genetic engineering of various species. These molecular scissors are able to perform a targeted gene modification through one of two major mechanisms, non-homologous end-joining, which results in a base pair alteration mutation over the break site, or homologous recombination, which drives gene insertion or gene replacement to the targeted locus [135,136]. The new approaches may be used not only to optimize specific characteristics of certain microalgal species, but also to develop novel traits into existing microalgal systems to fulfill industrial necessities [137].

All these techniques and approaches of genetic engineering of microalgae show their effectiveness to boost the production of high-value compounds. It is possible and necessary to extrapolate the knowledge base on cyanobacterial genetic engineering to the eukaryotic microalgal genetic engineering [133]. Though still challenging, increasingly more newly developed genetic tools and methods have been applied to enhance the transformation and expression systems such as removing the cell wall to improve the transformation efficiency [133]. Adjustment of the cultivation condition may result in great influences on the content of some biocompounds in microalgae. With a suitable and adequate association of genetic manipulation and culture process, optimal productivity could be achieved [133].

There are various key obstacles to overcome in microalgal genetic editing. The design and regulation of multiple target sites are primarily challenging and are then followed by the establishment of intact metabolic pathways and post-translocation stability. Sufficient genomic and transcriptomic data and information are essential in achieving the above goals. The low transformation efficiency requires pretreatments to eliminate cell walls or the use of a cell wall-deficient mutant deficient mutant [133]. In addition, label-free genetic editing might be instrumental for industrial application due to safety and regulatory reasons. To counter these limitations, the fundamental study of molecular elements, such as identification and cloning of promoters, enhancers and terminator, should be carried out more intensively. The innovation and toolkits for genetic engineering of microalgae are also needed to be specifically improved.

The main challenges appear to be the high cost of operation, infrastructure, maintenance, mass production, bioproduct accumulation and extraction [138]. Using industrial or agricultural waste contained with less microbial load should be adapted to a medium for sustainability and thus saving costs for the industrial scale. Indeed, fundamental knowledge and research are also necessary, making more research on various cultivation conditions a good option within the next few years. Many plant chemicals that are of pharmaceutical interest are waiting to be produced by the benefits of genetic engineering of microbial synthesis on an industrial scale. In terms of sustainability, combined with economic, environmental and short life cycle benefits, hetero- and autotrophic microalgae may reach this goal [139]. The lower biomass productivity, harvesting technology and poorly developed downstream biorefinery are the major reasons of high production cost of the bioproducts. With the improvement in each process step in microalgal cultivation, substantial progress towards a cost-efficient microalgal-based biorefinery technology needs to be developed [140]. The challenges to meet the economic demand are multifaceted, including both product quality and cost effectiveness. Improving yield and product quality in some microalgal hosts remains to be addressed [139]. Although a small number of microalgal hosts are approaching commercialization, as the demand for therapeutics and other industries is continually growing, there are still some limitations from genetic engineering of microalgal hosts, such as difficult engineering due to the lack of a high-efficiency genetic toolbox, less-available molecular specific toolkits, short-term stability of the genetic system and less efficient manipulation outside laboratory [139].

Safety is a primary consideration for microalgal products for human food and animal feed applications, and environmental safety associated with the cultivation of wild-type microalgae and genetically modified (GM) microalgal strains is also of great concern [140]. It is important that the food security agencies establish the permissible guidelines of GM-microalgal biomass for human and animal consumption, and furthering the potential of genetically modified microalgae needs to be assessed to ensure successful commercialization in order to support future energy and food security without risk to human and environmental health. The key safety legislation must be established to evaluate the environmental risk assessment (ERA) system that may be able to respond to rapidly evolving research and development [140].

## 4. Conclusions

In order to fulfill requirements of the world’s growing population, inexpensive as well as environmentally friendly production processes, including exploitation of microalgae, should be more aggressively developed. Although development of genetic engineering technologies makes it possible for researchers to engineer and improve production efficiency of microalgae, only a few models species were chosen to be used so far. With advances made in sequencing technologies as well as the availability of genomic data for a variety of microalgae strains, these quickly emerging technologies allow researchers to investigate and explore the wide range of this valuable group of organisms for commercial applications. In addition, synthesis genes are becoming available and applicable to modify algal strains, resulting in both time-saving and cost-effective genetic engineering procedures. Finally, in parallel and with all these technological advances, thorough studies must be carried out in order to evaluate and understand the effects of these genetically modified strains on the environment. The stability of these strains on a large scale is primordial before their use in industrial applications.

## Figures and Tables

**Figure 1 marinedrugs-20-00285-f001:**
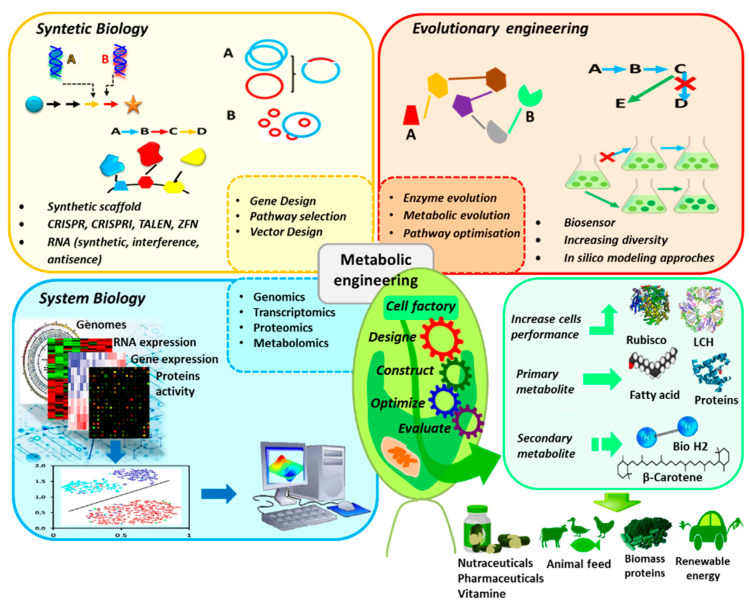
Genetic technology applied to microalgae to optimize production of target metabolites.

**Figure 2 marinedrugs-20-00285-f002:**
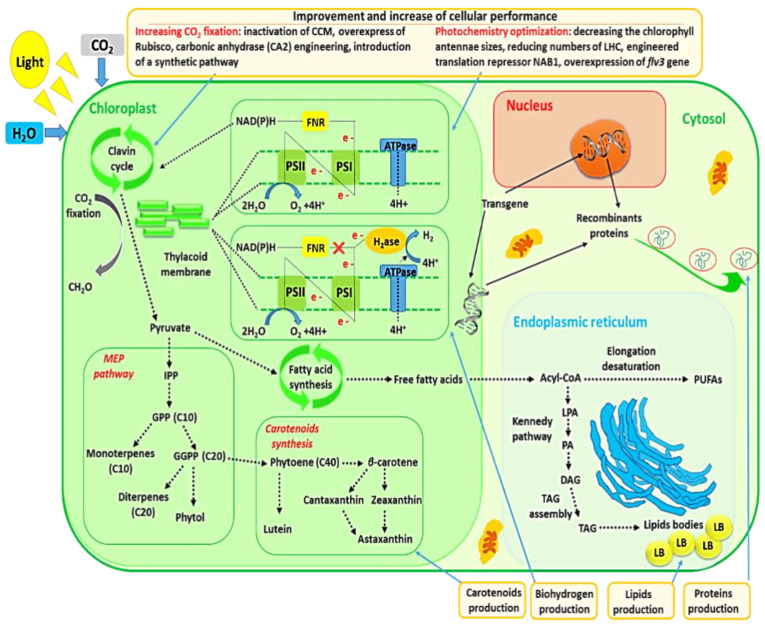
Commercial production of microalgal metabolites entangled with CO_2_ sequestration and photochemistry improvement.

**Table 3 marinedrugs-20-00285-t003:** Examples of genetic engineering of microalgae for carotenoid production.

Microalgae Strain	Gene/Target Site	Approach	Results	References
*Chlamydomonas* sp. JSC4	*lut1* and *zep*	High-intensity light induced repression of lut1 and zep	High lutein productivity (5.08 mg/L/d)	Ma et al. [76]
*Haematococcus pluvialis*	Endogenous phytoenedesaturase (*PDS*)	Codon optimized/ overexpressed	Accumulation of astaxanthin up to 67% higher	Galarza et al. [83]
*Chlamydomonas reinhardtii*	Zeaxanthin epoxidase (*ZEP*)	DNA-free CRISPR-Cas9, knock-out mutant	Increase in both zeaxanthin content and productivity by 56- and 47-fold, respectively	Baek et al. [81]
*Haematococcus pluvialis*	*β*-carotene ketolase (*bkt*)	Cloning and overexpressed	Increase in total carotenoids and astaxanthin content by 2–3-fold higher	Kathiresan et al. [79]
*Haematococcus pluvialis*	*HpDGAT1*	Upregulated expression	Increase in esterified astaxanthin (EAST)	Cui et al. [84]
*Haematococcus pluvialis*	*β*-carotene ketolase and b-carotene hydroxylase	Cloning and expression plasmids’ construction	Genes *PSY*, *PDS*, *ZDS*, *LCYB* expressed 2~4 fold higher, with amount of astaxanthin of 5.56 mg/g dry weight	Chen et al. [85]
*Chlamydomonas reinhardtii*	*β*-carotene ketolase (*Cr*BKT)	Overexpression of the optimized *Cr*BKT	Up to 50% of native carotenoids could be converted into astaxanthin	Perozeni et al. [86]
*Phaeodactylum tricornutum*	*dxs* and *psy*	Transcriptional upregulation	2.4-fold and a 1.8-fold higher fucoxanthin content, respectively	Eilers et al. [87]
*Phaeodactylum tricornutum*	Phytoene synthase gene (*psy*)	Transformation and gene expression	Increased the fucoxanthin content by approximately 1.45-fold	Kadono et al. [88]
*Dunaliella tertiolecta*	Carotenoid biosynthesis-related (*CBR*)	Antisense expression and overexpression	Zeaxanthin increased with the increasing irradiation time by 2.22-fold	Zhang et al. [89]
*Chlamydomonas reinhardtii*	Bifunctional *PBS* gene	Heterologous expression	38% enhancement in *β*-carotene along with 60% increase in the lutein	Rathod et al. [90]
*Chlamydomonas reinhardtii*	*DXS* and *DXR*	Overexpressed via nuclear transformation	Increased lutein and *β*-carotene by 1.9-fold and 1.7-fold per cell, respectively	Morikawa et al. [91]
*Dunaliella salina*	Introduction of a *bkt* gene	Transformation procedure	Astaxanthin and canthaxanthin with maximum content of 3.5 and 1.9 lg/g DW, respectively	Anila et al. [78]
*Dunaliella tertiolecta*	*mp3*	Random mutagenesis	10–15% higher cellular zeaxanthin content	Kim et al. [82]
*Chlorella zofingiensis*	Phytoene desaturase (*PDS*)	Overexpression	Increase total carotenoid and astaxanthin production by 32.1% and 54.1% respectively.	Liu et al. [77]

## Data Availability

Not applicable.

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
