# Peer review of "Emerging Trends in Genetic Engineering of Microalgae for Commercial Applications"

_marinedrugs, 2022, doi:10.3390/md20050285_

Round 1

Reviewer 1 Report

The present manuscript comprehensively discusses recent research advances on microalgae genetic engineering. I have, however, suggestions that need to be addressed.

The authors use inconsistent terminology. They use microalgae or algae. Use preferably microalgae since algae also include seaweeds. Also, be clear with the distinction of microalgae and cyanobacteria; they are not the same. In the introduction, the difference between microalgae and cyanobacteria should be stated. The article seems to focus too much on microalgae and not cyanobacteria.

Rephrase the first paragraph of the introduction. It is not clearly redacted.

In the second paragraph, look for a more recent publication about the number of microalgal and cyanobacterial species. E.g.,  the work of Guiry et al. (2012) entitled “How many species of algae are there?“

In section 2 first paragraph. Change the order of the sentence. Biohydrogen is not a pharmaceutical product but a fuel and therefore should be listed after carotenoids.

It is important to include in the manuscript not only the increase in a metabolite but also how the genetic modifications have affected or not the cell growth. This discussion is missing in each section. In addition, there needs to be a more specific comparison of the improvements based on genetic engineering in each section. This is well stated, for instance, for hydrogen but needs to be improved for the rest of the metabolites. Each section needs to end with a small summary (take-home lesson) that the authors want to give to the reader.

In section 2.6, the reference is missing after “ 200 – 400 umol/m2s for the majority of the microalgal species.”

I disagree with the conclusion section; this is part of the discussion. Include it in the manuscript as another section and redact the conclusions.

Also, improve the abstract. It is written as an introduction—too general information.

Author Response

Reviewer #1:  The present manuscript comprehensively discusses recent research advances on microalgae genetic engineering. I have, however, suggestions that need to be addressed.

Our response: We appreciate your reviewing efforts and suggestions improving our manuscript.

Reviewer #1:  The authors use inconsistent terminology. They use microalgae or algae. Use preferably microalgae since algae also include seaweeds. Also, be clear with the distinction of microalgae and cyanobacteria; they are not the same. In the introduction, the difference between microalgae and cyanobacteria should be stated. The article seems to focus too much on microalgae and not cyanobacteria.

Our response: Thanks a lot for the clarification. We modified the manuscript to eliminate ambiguities between microalgae and algae, which could confuse the reader. And we agree with you that our manuscript focus mostly on microalgae instead of caynobacteria, which we in fact intend to attribute them also as microalgae.

Reviewer #1:  Rephrase the first paragraph of the introduction. It is not clearly redacted.

Our response: We agree that the first paragraph was a little bit awkward. We have tried our best to modify it.  

Reviewer #1:  In the second paragraph, look for a more recent publication about the number of microalgal and cyanobacterial species. E.g.,  the work of Guiry et al. (2012) entitled “How many species of algae are there?“

Our response: Thank you very much for letting us know this interesting references, and we updated our knowledge about algae and include it in our manuscript.

Reviewer #1:  In section 2 first paragraph. Change the order of the sentence. Biohydrogen is not a pharmaceutical product but a fuel and therefore should be listed after carotenoids.

Our response: Modified.

Reviewer #1:  It is important to include in the manuscript not only the increase in a metabolite but also how the genetic modifications have affected or not the cell growth. This discussion is missing in each section.

Our response: We agree that you have a valid and relevant point concerning the addition of the growth rate impact. This is now included in the document. Growth rate or influence on growth was added in the works that mentioned this parameter. Other works cited did not mention this parameter in their studies.

Reviewer #1:  In addition, there needs to be a more specific comparison of the improvements based on genetic engineering in each section. This is well stated, for instance, for hydrogen but needs to be improved for the rest of the metabolites.

Our response: A very good suggestion. We have tried to improve the manuscript for the rest of the metabolites.

Reviewer #1:  Each section needs to end with a small summary (take-home lesson) that the authors want to give to the reader.

Our response: A  very good suggestion. We have tried to add more comments to the mannuscript.

Reviewer #1:  In section 2.6, the reference is missing after “ 200 – 400 umol/m2s for the majority of the microalgal species.”

Our response: Corrected.

Reviewer #1:  I disagree with the conclusion section; this is part of the discussion. Include it in the manuscript as another section and redact the conclusions.

Our response: Thanks for the careful review. It is corrected. Another paragraph entitled, challanges and prospects has been added before the conclusion. Also this last one has been improved to be more clear

Reviewer #1:  Also, improve the abstract. It is written as an introduction—too general information.

Our response: Modified.

Reviewer 2 Report

Comments for authors consideration are as follows:

  1. The given abstract does not deduce the justification of the work theme.
  2. Introduction is too little to spot the background. It must be extended by adding recent aspects on industrial feasibility, socioeconomic considerations and so on.
  3. Table 1 is superficial. More recent studies should be added to further extend with at least 15-20 references.
  4. Tables 2, 3 and 4, same comment as above for Table 1.
  5. More Figures should be added.
  6. A separate section must be added by detailing complications of genetic engineering of microalgae for commercial applications.
  7. Before conclusion, add a section on challenges. What are authors own viewpoints? What are the major findings and how they are addressing the left behind research gaps and current challenges?

Author Response

Reviewer #2:

  1. The given abstract does not deduce the justification of the work theme.

Thanks a lot for the critical comment. We tried our best to modify the abstract, and please evaluate it again and give us more guidance on it.

  1. Introduction is too little to spot the background. It must be extended by adding recent aspects on industrial feasibility, socioeconomic considerations and so on.

We definitely agree with you on this point, but the time period is too limited for us to be able to fully address it.

  1. Table 1 is superficial. More recent studies should be added to further extend with at least 15-20 references.

Thanks a lot for mentioning this fact. We have modified our manuscript, adding more cases to the tables.

  1. Tables 2, 3 and 4, same comment as above for Table 1.

We have modified our manuscript, for all three tables, proteins, lipids and carotenoids.

  1. More Figures should be added.

We agree. We are working on one more figure, and we hope to be able to add it in the final draft.  

  1. A separate section must be added by detailing complications of genetic engineering of microalgae for commercial applications.

We believe this is a very good suggestion. A new section entitled, challanges, opportunities and prospects has been added before the conclusion.

  1. Before conclusion, add a section on challenges.

 A new section entitled, challanges, opportunities and prospects has been added before the conclusion.

  1. What are authors own viewpoints? What are the major findings and how they are addressing the left behind research gaps and current challenges?

Thank you very much for your insightful comments. We have tried to fit in as much as we can your suggested contents into the manuscript although we have too short period of time  to response your valued suggestions and comments.

Round 2

Reviewer 1 Report

The changes performed by the authors are acceptable.  

I only have one comment. Microalgae are not plants. Modify in the introduction (first paragraph).

Author Response

We modified the manuscript as instructed. Thanks a lot!

Reviewer 2 Report

The revised version reads well. Authors have addressed all the comments raised in the last review. This manuscript can now be accepted for publication.

Author Response

Thanks a lot for your efforts and endorsement of our manuscript.